# Desmoplastic Small Round Cell Tumor of the Uterus: A Report of Molecularly Confirmed Case with EWSR1-WT1 Fusion

**DOI:** 10.3390/diagnostics12051184

**Published:** 2022-05-10

**Authors:** Pavel Dundr, Jana Drozenová, Radoslav Matěj, Michaela Bártů, Kristýna Němejcová, Helena Robová, Lukáš Rob, Jan Hojný, Ivana Stružinská

**Affiliations:** 1Department of Pathology, First Faculty of Medicine, General University Hospital in Prague, Charles University, 12808 Prague, Czech Republic; radoslav.matej@ftn.cz (R.M.); michaela.bartu@vfn.cz (M.B.); kristyna.nemejcova@vfn.cz (K.N.); jan.hojny@vfn.cz (J.H.); ivana.struzinska@vfn.cz (I.S.); 2Department of Pathology, Third Faculty of Medicine, University Hospital Kralovske Vinohrady, Charles University, 10034 Prague, Czech Republic; jana.drozenova@fnkv.cz; 3Department of Pathology and Molecular Medicine, Third Faculty of Medicine, Thomayer Hospital, Charles University, 14059 Prague, Czech Republic; 4Department of Obstetrics and Gynecology, Third Faculty of Medicine, University Hospital Kralovske Vinohrady, Charles University, 10034 Prague, Czech Republic; helena.robova@fnkv.cz (H.R.); lukas.rob@fnkv.cz (L.R.)

**Keywords:** desmoplastic small round cell tumor, uterus, EWSR1-WT1 fusion

## Abstract

We report a case of a 49-year-old female with desmoplastic small round cell tumor of the uterus (DSRCT). Histologically, in some areas the tumor showed typical features with ample desmoplastic stroma, while in other areas the tumor cells diffusely infiltrated myometrium with only focal desmoplastic reaction. Immunohistochemically, the tumor cells showed diffuse positivity for desmin, CD56, CD57, EMA and cyclin D1. Focal positivity was present for antibodies against cytokeratin AE1/3, BerEP4, NSE, IFITM1 and CD10. The WT-1 antibody (against the N-terminus) showed cytoplasmic positivity in some tumor cells, while the nuclei were negative. P53 expression was wild-type. The Ki-67 index (MIB1 antibody) was about 55%. Other markers examined including transgelin, myogenin, synaptophysin, chromogranin, h-caldesmon, PAX8, and CD117 were all negative. NGS analysis revealed a fusion transcript of the EWSR1 and WT1 genes. DSRCT of the uterus is a rare neoplasm, as only two cases have been reported so far. However, only one of these cases was examined molecularly with a confirmation of the characteristic EWSR1-WT1 fusion. We report a second case of molecularly confirmed DSRCT of the uterus and discuss its clinical features, differential diagnosis and the significance of molecular testing.

## 1. Introduction

Desmoplastic small round cell tumor (DSRCT) is an aggressive neoplasm, usually arising in the abdominal and/or pelvic region in children and young adults, with most cases (~85%) occurring in male patients [1]. However, rare cases originating in other locations have also been described, including the head and neck area, the liver, pleura, soft tissues, kidney, the inguinal/paratesticular area, as well as intracranial and intraosseous cases [2,3,4]. In the uterus, only two cases of DSCRT have been reported so far [2,5]. However, only one of these cases was molecularly examined with a confirmation of the characteristic EWSR1-WT1 fusion [2]. In the second case, the diagnosis was based on the morphology and immunohistochemical findings alone [5]. A few other cases of DSRCT have been described in the female genital tract, one of which arose in the vagina while the rest were found in the ovary [6,7,8]. In this work, we report on the morphological, immunohistochemical, and molecular features of another case of uterine DSRCT arising in the uterus of a 49-year-old female.

## 2. Case Presentation

### 2.1. Clinical Findings

A 49-year-old woman was referred to the gynecological department of our institution following the identification of a tumor located in the lower uterine segment, which was growing exophytically into the cervical canal. There was no history of previous pelvic irradiation. The patient underwent a biopsy in a regional hospital with inconclusive findings of “mesenchymal tumor” without specific diagnosis. The ultrasonography, CT scan, and MRI showed a heterogeneous tumor mass of 73 mm × 49 mm × 44 mm located in the lower uterine segment and infiltrating the uterine wall and cervix (Figure 1 and Figure 2). The imaging methods were also suggestive of pelvic lymphadenopathy. A repeated biopsy of the portion of the tumor which protruded into the cervix showed a malignant, probably mesenchymal, tumor which was, however, difficult to classify due to the sample limitations (necrosis, limited amount of viable tumor cells). A CT scan of the chest was negative for metastases at the time of diagnosis. Based on the results of the biopsy, the patient underwent a radical hysterectomy with bilateral salpingo-oophorectomy, combined with pelvic and suprapelvic lymphadenectomy in December 2020. After the surgery, the patient underwent two cycles of adjuvant chemotherapy (cisplatin and doxorubicin), followed by pelvic and para-aortic radiotherapy and additional two cycles of chemotherapy (cisplatin and doxorubicin). The post-operative PET-CT scan showed no residual disease. After 14 months of follow-up, the patient is alive with no evidence of disease.

### 2.2. Morphological Findings

The resection specimen consisted of the uterus with the vaginal cuff and both adnexa. The lower uterine segment was occupied by a tumor mass of 72 mm × 48 mm × 35 mm, with an infiltration into the uterine cervix. The tumor distance from vagina was 8 mm. The tumor had invaded into more than half of the myometrium thickness, but without extension to the serosa. The lymphadenectomy specimens consisted of 23 lymph nodes. Microscopically, the tumor consisted of medium-sized round or oval tumor cells (Figure 3A–C). The chromatin was coarse, mostly with non-prominent nucleoli. Mitotic figures were numerous (up to 21/10 HPF). The cytoplasm was scant, eosinophilic or amphophilic, and rare tumor cells displayed a rhabdoid appearance. A substantial invasion into vascular spaces was present. Small areas of coagulative necrosis were found. The neoplastic cells in some areas had formed well-demarcated nests, sheets, and trabeculae located in ample desmoplastic (mostly hypocellular) stroma. In other areas the tumor cells diffusely infiltrated myometrium with only focal desmoplastic reaction. In these areas, the tumor cell formed irregular groups, nests, trabeculae, and sheets, or dissociated in single cells or small groups. Tumor metastases were found in 15 lymph nodes (regional and para-aortic) from the 23 examined. Both the adnexa and vaginal cuff were without pathological changes.

### 2.3. Immunohistochemical Findings

The immunohistochemical (IHC) analysis was performed using 4 μm thick sections of formalin-fixed and paraffin-embedded (FFPE) tissue. The list of antibodies used, including their clones, manufacturers, dilution, and staining instruments is summarized in Appendix A. The tumor cells showed diffuse positivity for desmin (Figure 3D), CD56, CD57, EMA and cyclin D1 (strong nuclear). Focal positivity was present for antibodies against cytokeratin AE1/3 (Figure 3E), BerEP4, NSE, IFITM1 and CD10. The WT-1 antibody (against the N-terminus) showed cytoplasmic positivity in some tumor cells, while the nuclei were negative (Figure 3F). P53 expression was wild-type. The Ki-67 index (MIB1 antibody) was about 55%. Other markers examined, including transgelin, myogenin, synaptophysin, chromogranin, h-caldesmon, PAX8 and CD117, were negative.

### 2.4. Molecular Findings

The RNA NGS analysis to detect gene fusions was performed as described previously using Archer FusionPlex Sarcoma Expanded Kit (ArcherDX, Boulder, CO, USA) and Archer Analysis software v5.1.7 (ArcherDX, Boulder, CO, USA) according to the manufacturer’s instructions [9]. Detailed pipelines of all NGS data analysis together with module settings are available upon request. The analysis revealed a fusion transcript of the EWSR1 (exon 11; NM_013986.3) and WT1 (exon 8; NM_024424.3) genes. The fusion was detected in 3923 unique reads (65.1% of the reads spanning the breakpoint). The detected fusion maintains an open reading frame and does not create a premature termination codon.

## 3. Discussion

DSRCT is an aggressive tumor with characteristic morphological, immunohistochemical and molecular features [1,10]. Morphologically, the characteristic features mainly include the arrangement of the tumor cells into sharply outlined groups or nests of variable shape and size, which are set in desmoplastic stroma. The stroma can vary in amount and cellularity. In typical cases, the tumor consists of oval cells with a high nuclear-to-cytoplasm ratio, which can also show rhabdoid features or clear cytoplasm resulting in a signet-ring cell appearance in a minority of cases [1,3]. Mitotic figures are usually numerous. However, some cases can have an unusual morphology including tubular structures, pseudorosette formation, squamous component, and spindle cell morphology [3,11]. The immunohistochemical profile of DSRCT is characterized by the co-expression of keratins, EMA, desmin, and WT1 [12]. However, one should be aware that the typical nuclear WT1 positivity is present only when using antibodies against the C-terminus, as the antibodies against the N-terminus are negative in tumor cell nuclei (although they can show non-specific cytoplasmic positivity) [13]. Other markers can be expressed, including neuroendocrine markers, actin and caldesmon, but this expression is usually only focal and found in rare cases [12]. In cases with a typical morphology, the diagnosis is usually straightforward, but in cases with an unusual morphology or in cases arising in atypical locations, the process can be problematic. In such cases, molecular findings can be helpful, because all DSRCT cases should possess a balanced reciprocal translocation t(11;22)(p13;q12), resulting in a fusion of the EWSR1 and WT1 genes (EWSR1-WT1). This fusion used to be regarded as a specific molecular aberration occurring only in DSRCT [14]; however, there have been reports of a total of six tumors with EWSR1-WT1 fusion which showed morphological and immunohistochemical features different from DSRCT [15,16,17]. These tumors were found in the vagina, uterine cervix, cauda equina, small intestine, pelvis, and one of them presented multifocally (uterus, ovaries, retroperitoneum, peritoneum, and pelvis). Morphologically, two of these tumors consisted of bland ovoid to spindle cells, one case had features of monomorphic spindle cell sarcoma, two cases resembled leiomyosarcoma with spindled and focally epithelioid cells, and the last case (located in cauda equina) consisted of cells with glomoid features, forming rosette-like structures located in hyalinized and myxoid stroma. No case showed the desmoplasia typical for DSRCT. Immunohistochemically, all cases were positive for desmin and smooth muscle actin (SMA), five cases were positive for keratins (two of them only focally), and two out of five cases showed focal positivity for h-caldesmon. From the five cases in which WT1 (with antibody against the C-terminus) was examined, all showed nuclear positivity. All six cases were also examined with the WT1 antibody against the N-terminus and, with the exception of one case (which showed focal cytoplasmic positivity), all cases were negative.

The differential diagnosis of DSRCT arising in the uterus includes, especially, primitive neuroectodermal tumor (PNET), CIC-rearranged sarcomas, undifferentiated endometrial carcinoma (UDEC), SMARCA4-deficient uterine sarcoma (SDUS), high grade endometrial stromal sarcoma (HG-ESS), and neuroendocrine carcinoma. Although these tumors are generally not characterized by the desmoplasia which is characteristic for DSRCT, the typical morphological pattern is not necessarily present in all DSRCTs—in our case, the typical morphology was present only in some (minor) areas of the tumor, which is why other features should be assessed in the differential diagnosis of these tumors as well.

Uterine PNET can be of either the central or peripheral type [18,19,20]. The central type of uterine PNET consists of small round blue cells with possible neuronal or glial differentiation. Unlike the peripheral type, the central type PNET lacks EWSR1 rearrangement. This feature can also be helpful in distinguishing it from DSRCT. Compared to DSRCT, the immunohistochemical findings of the central type of PNET are different. The tumor is often positive for chromogranin, synaptophysin, S100 protein, and in about 50% of cases for GFAP. CD99 and FLI1 can be positive, but desmin is typically negative. The peripheral type of PNET (which is synonymous with Ewing sarcoma) is rare in the uterus, and its distinction from DSRCT may be problematic. Nevertheless, similarly to the central type of PNET, the immunohistochemical profile differs from DSRCT. Moreover, the molecular aberrations found in peripheral PNET (Ewing sarcoma) are different from the EWRS1-WT1 fusion typical for DSRCT. The most common fusion occurring in Ewing sarcoma is EWRS1-FLI1, followed by EWSR1-ERG and less common aberrations including fusion of EWSR1 with other members of the ETS family, or fusion of the FUS gene (which is a member of FET family, together with EWSR1 and TAF15) with certain genes from the ETS family [21]. Other tumors closely resembling Ewing sarcoma which should be considered in the differential diagnosis of DSRCT are tumors with the rearrangement of CIC, most often resulting in CIC-DUX4 fusion [22]. These tumors were formerly classified as Ewing-like sarcomas but are currently regarded as distinct entities with highly aggressive behavior and a substantially worse prognosis than Ewing sarcoma. These tumors mostly occur in soft tissues, but can arise in the bones, GIT, kidney, prostate, and the head and neck region. Recently, two reports were published describing a sarcoma with CIC-DUX4 fusion located in the uterine corpus and uterine cervix [23,24]. Immunohistochemically, CIC-rearranged sarcomas are usually positive for CD99 (84%) and WT1 (92%) [22]. However, in contrast to DSRCT, keratin is expressed in only 15% of cases and desmin in 3.6% of cases.

Other tumors which should be considered in the differential diagnosis are UDEC and SDUS, which have some overlapping features [25]. These tumors consist of oval cells with a high nuclear-to-cytoplasm ratio and increased mitotic activity, which can have rhabdoid morphology. Stromal hyalinization can be present, especially in SDUS. Immunohistochemically, these tumors can focally express keratins, 100% of SDUS and 20–30% of UDEC have a loss of SMARCA4 expression, about 30% of UDEC show aberrant p53 expression, and about 50% are mismatch repair (MMR) protein deficient. SDUS is MMR proficient and a p53 wild-type. The immunohistochemistry can be helpful in differential diagnosis, but the significance of some of these markers for differential diagnosis with DSRCT is not known and molecular testing may be needed due to the possible morphological variability of these tumors.

Differential diagnosis between DSRCT and HG-ESS can be more problematic due to the heterogeneity of HG-ESS. Some HG-ESS are morphologically, immunohistochemically, and molecularly well characterized, while others probably represent heterogeneous entities, and their classification has been evolving rapidly over recent years. From the tumors which are currently well characterized, the differential diagnosis includes especially YWHAE-FAM22(NUTM2) translocated HG-ESS [26]. Similarly to DSRCT, this tumor consists of oval cells, but their arrangement is different with a growth pattern similar to low grade endometrial stromal sarcoma with tongue-like myometrial infiltration. Desmoplasia is not a typical feature of YWHAE-FAM22(NUTM2) translocated HG-ESS [26,27]. Other HG-ESS (such as BCOR and BCORL1 altered) and uterine sarcoma with kinase fusions are characterized by spindle cells, and their morphological features are different [28,29,30].

The differential diagnosis with poorly differentiated neuroendocrine carcinoma can be challenging in some cases, especially since DSRCT typically express cytokeratins, and rarely they can also show expression of neuroendocrine markers such as synaptophysin, chromogranin, and INSM1 [31]. Nevertheless, other markers, such as desmin and WT1, are not expressed in neuroendocrine tumors.

Finally, other tumors which could be considered in broader differential diagnosis are carcinosarcoma and lymphoma. These tumors should be excluded based on either their morphological features (epithelial component in carcinosarcoma), or their different immunohistochemical profile (lymphoma).

## 4. Conclusions

We have described a case of DSRCT arising in the uterus with typical morphology, immunohistochemical features, and the presence of characteristic EWSR1-WT1 fusion. This entity should be kept in mind when considering the differential diagnosis of poorly differentiated round cell tumors of the uterus. The molecular characterization of the tumor with detection of EWSR1-WT1 fusion may be essential in confirming the diagnosis and differentiating DSRCT from tumors showing overlapping morphologic and immunohistochemical features. However, according to the literature, the EWSR1-WT1 fusion does not seem to be specific exclusively for DSRCT and can occur in some other tumors as well, including mesenchymal tumors of the uterus. The presence of EWSR1-WT1 fusion is therefore not diagnostic for DSRCT per se. The other entities with EWSR1-WT1 fusions seem to represent a heterogeneous group of lesions showing different morphological features from those typical for DSRCT. They probably do not represent a distinct diagnostic entity, but further research on this topic is needed. Due to the rapidly evolving molecular classification (especially when it comes to mesenchymal uterine tumors) and the overlapping morphological and immunohistochemical features of certain entities which can be included in the differential diagnosis with DSRCT, molecular testing should be a standard part of the diagnostic process of these unusual tumors.

## Figures and Tables

**Figure 1 diagnostics-12-01184-f001:**
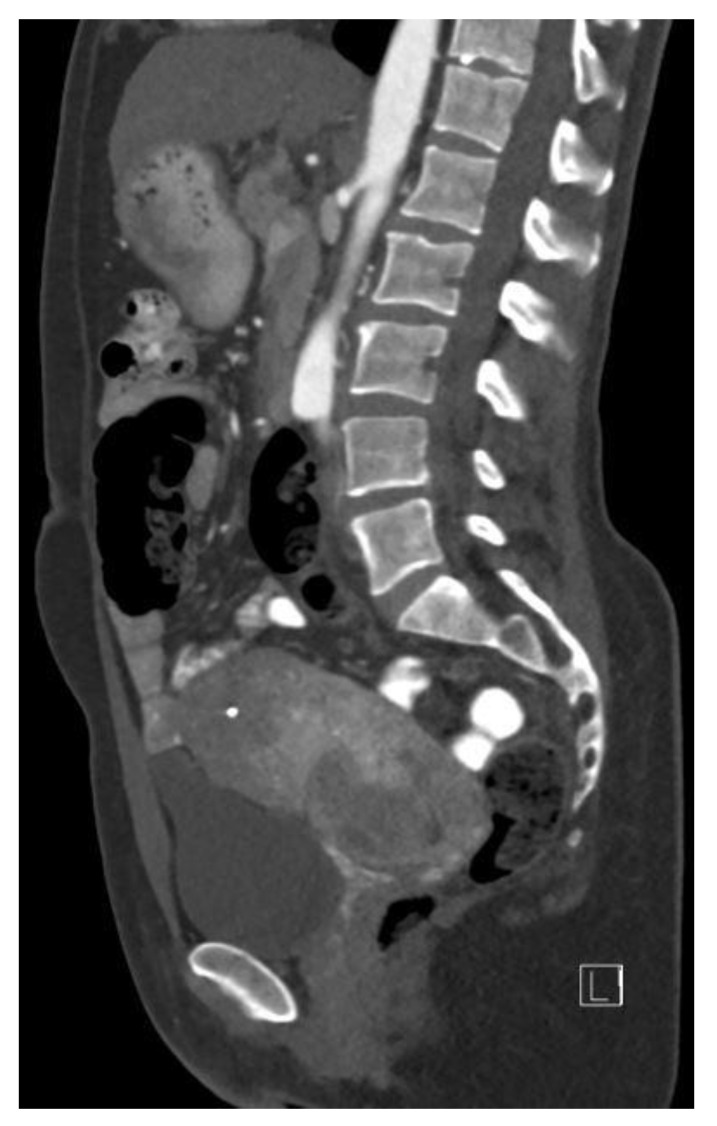
Computerized tomography (CT). Sagittal section of the cervical region. The tumor dimensions are 70 mm × 52 mm × 55 mm, and it is located in the lower uterine segment infiltrating the cervix and vaginal vault. The tumor is of heterogenous density, richly vascularized, and sharply demarcated with suspected infiltration of the parametria. There is also an intrauterine device in situ. Pelvic lymphadenopathy present.

**Figure 2 diagnostics-12-01184-f002:**
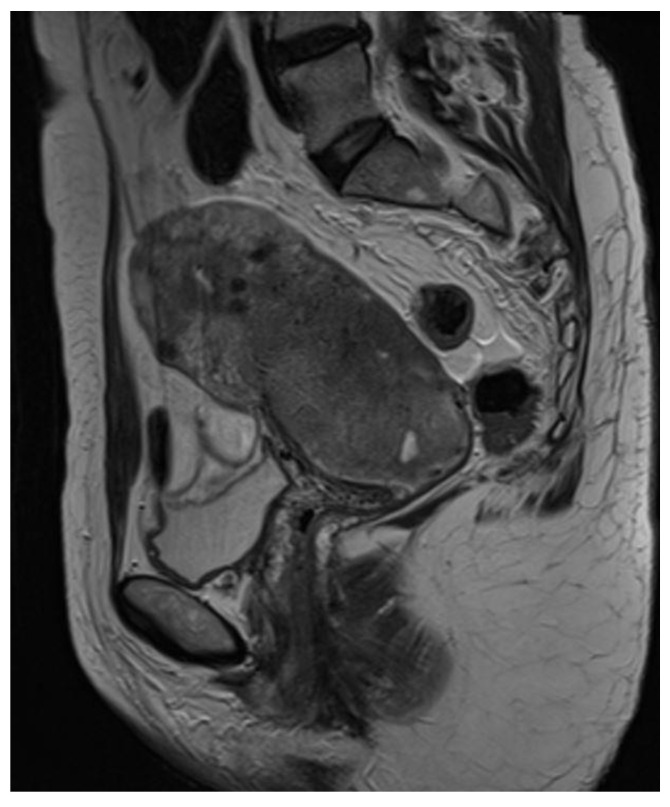
Magnetic resonance (MRI, T2)—sagittal section. Hypersignal tumor 73 mm × 44 mm × 49 mm with apparent neovascularization in the lower uterine body and uterine cervix. The MRI image is suspicious for a sarcomatoid tumor. Intrauterine device in situ. Pelvic lymphadenopathy present.

**Figure 3 diagnostics-12-01184-f003:**
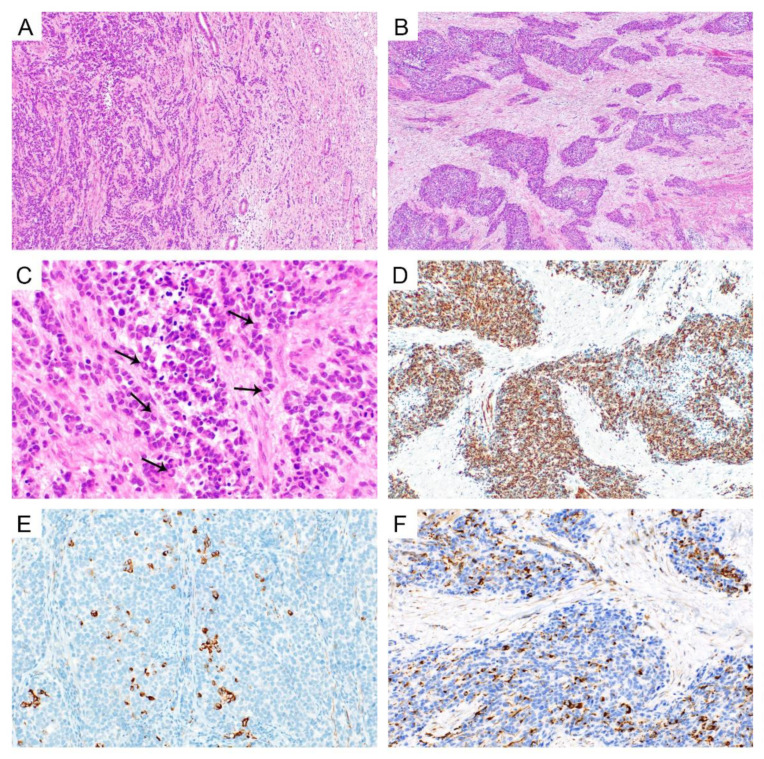
Desmoplastic small round cell tumor. Tumor infiltrating the lower uterine segment and upper endocervix. Note the residual mucosa on the right (**A**) (HE, 100×). Groups of tumor cells separated by dense fibroblastic stroma (**B**) (HE, 40×). High magnification showing undifferentiated tumor cells, some of them with rhabdoid features (arrows) (**C**) (HE, 400×). Immunohistochemical findings with desmin positivity in most tumor cells (**D**) (100×), focal positivity of cytokeratin AE1/3 (**E**) (200×), and non-specific cytoplasmic WT1 positivity (**F**) (200×).

## Data Availability

NGS raw data are available on request.

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
