# Peer review of "Desmoplastic Small Round Cell Tumor of the Uterus: A Report of Molecularly Confirmed Case with EWSR1-WT1 Fusion"

_diagnostics, 2022, doi:10.3390/diagnostics12051184_

Round 1

Reviewer 1 Report

Dear Authors,
I read your manuscript with interest and curiosity, which I found interesting. This is the description of a single case of a rare variant of mesenchymal neoplasm such as round blue small cell desmoplastic tumor (DSRCT) at the level of a very rare localization such as the uterus. The paper is well written and conversational.
Here are some comments to improve the quality:
Minor comments:

Figure 1 is of good resolution but needs to be improved: first of all, I would suggest to the authors to replace Figure 1A with a figure that shows both the DSCRT and the histological characteristics of a uterus. Figure 1B can be enhanced by inserting black arrows to indicate cells with rhabdoid differentiation. Immunostaining figures are fine. Please correct the enlargements of the figures: either all multiplying by 10 (for example 400x), or all without multiplication (for example 40x).

Please, cite figure 1 in the main text.

Reference style is not adequate for MDPI journals. Please, correct!

Author Response

Dear reviewer,

We would like to thank you for the evaluation, valuable comments, and suggestions which hopefully have led to the improvement of the manuscript. Our response to all reviewers’ comments is listed point-by-point below. All changes made in the revised manuscript are highlighted (colored) in tracking mode. The modified manuscript with highlighted changes has been uploaded via the submission page.

Dear Authors,
I read your manuscript with interest and curiosity, which I found interesting. This is the description of a single case of a rare variant of mesenchymal neoplasm such as round blue small cell desmoplastic tumor (DSRCT) at the level of a very rare localization such as the uterus. The paper is well written and conversational.
Here are some comments to improve the quality:
Minor comments:

Figure 1 is of good resolution but needs to be improved: first of all, I would suggest to the authors to replace Figure 1A with a figure that shows both the DSCRT and the histological characteristics of a uterus. Figure 1B can be enhanced by inserting black arrows to indicate cells with rhabdoid differentiation. Immunostaining figures are fine. Please correct the enlargements of the figures: either all multiplying by 10 (for example 400x), or all without multiplication (for example 40x).

Response: Thank you very much for this comment. We have changed the figure as follows: i) new HE figure (1B) was added, showing the tumor infiltration of the lower uterine segement / upper endocervix, with apparent residual non-tumor mucosa; ii) we have inserted black arrows pointing at the rhabdoid cells; iii) we deleted the IHC image showing the results of EMA staining (this figure was replaced by the new HE figure); the magnification of all figures is multiplied by 10 (i.e. "40x" corresponds to low power view = 4x objective and 10x occular magnification; "400x" corresponds to high power view)

Please, cite figure 1 in the main text.

Response: Thank you very much for this comment. We have added the citation into the text.

Reference style is not adequate for MDPI journals. Please, correct!

Response: Thank you very much for this comment. We have changed it to be in accordance with the MDPI style.

Concerning the request for language correction, the article was referred back to the native speaker who performed the original language check (Mgr. Zachary Harold Kane Kendall, B.A. Institute for History of Medicine and Foreign Languages, First Faculty of Medicine, Charles University) and based on his expertise no changes are needed.

Reviewer 2 Report

(1) Had the patient any history of pelvic irradiation before this surgery?

(2) Could the authors show the pictures of CT scan and MRI?

(3) Could the authors show the pictures of surgical specimen?

(4) Please describe the rationale of choosing cisplatin and doxorubicin as the chemotherapeutic agents.

Author Response

Dear reviewer,

We would like to thank you for the evaluation, valuable comments, and suggestions which hopefully have led to the improvement of the manuscript. Our response to all reviewers’ comments is listed point-by-point below. All changes made in the revised manuscript are highlighted (colored) in tracking mode. The modified manuscript with highlighted changes has been uploaded via the submission page.

(1) Had the patient any history of pelvic irradiation before this surgery?

Response: Thank you for raising this issue. The patient has no history of pelvic irradiation. We have added this information into the manuscript.

(2) Could the authors show the pictures of CT scan and MRI?

Response: Thank you very much for this comment. We have added the representative CT and MRI into the manuscript.

(3) Could the authors show the pictures of surgical specimen?

Response: Thank you for this comment. Unfortunately, the figures of the gross features are not available, as the surgical specimen was not processed at our department.

 (4) Please describe the rationale of choosing cisplatin and doxorubicin as the chemotherapeutic agents.

Response: Thank you very much for this comment. This regimen was chosen based on the results of the discussion of the entire multidisciplinary team. The rationale is experience-based, as the tumor is rare and currently there is no standard chemotherapy regimen for these tumors in this age group.